# Dietary Plant Polysaccharides for Cancer Prevention: Role of Immune Cells and Gut Microbiota, Challenges and Perspectives

**DOI:** 10.3390/nu15133019

**Published:** 2023-07-03

**Authors:** Anqi Wang, Ying Liu, Shan Zeng, Yuanyuan Liu, Wei Li, Dingtao Wu, Xu Wu, Liang Zou, Huijuan Chen

**Affiliations:** 1School of Preclinical Medicine, Chengdu University, Chengdu 610106, China; wanganqi@cdu.edu.cn (A.W.); yingliu202112@163.com (Y.L.); liweivirlee@163.com (W.L.); 2Antibiotics Research and Re-Evaluation Key Laboratory of Sichuan Province, Sichuan Industrial Institute of Antibiotics, Chengdu University, Chengdu 610052, China; zengshan@126.com (S.Z.); liuyuanyuan02@126.com (Y.L.); 3Key Laboratory of Coarse Cereal Processing of Ministry of Agriculture and Rural Affairs, School of Food and Biological Engineering, Chengdu University, Chengdu 610106, China; wudingtao@cdu.edu.cn (D.W.); jianboxiao@yahoo.com (L.Z.); 4Laboratory of Molecular Pharmacology, Department of Pharmacology, School of Pharmacy, Southwest Medical University, Luzhou 646000, China; wuxulz@126.com; 5Institute of Traditional Chinese Medicine, Sichuan Academy of Chinese Medicine Sciences, Chengdu 610031, China

**Keywords:** dietary plant polysaccharides, cancer prevention, immune regulation, gut microbiota, chemical structure, immune cells

## Abstract

Dietary plant polysaccharides, one of the main sources of natural polysaccharides, possess significant cancer prevention activity and potential development value in the food and medicine fields. The anti-tumor mechanisms of plant polysaccharides are mainly elaborated from three perspectives: enhancing immunoregulation, inhibiting tumor cell growth and inhibiting tumor cell invasion and metastasis. The immune system plays a key role in cancer progression, and immunomodulation is considered a significant pathway for cancer prevention or treatment. Although much progress has been made in revealing the relationship between the cancer prevention activity of polysaccharides and immunoregulation, huge challenges are still met in the research and development of polysaccharides. Results suggest that certain polysaccharide types and glycosidic linkage forms significantly affect the biological activity of polysaccharides in immunoregulation. At present, the *in vitro* anti-tumor effects and immunoregulation of dietary polysaccharides are widely reported in articles; however, the anti-tumor effects and *in vivo* immunoregulation of dietary polysaccharides are still deserving of further investigation. In this paper, aspects of the mechanisms behind dietary polysaccharides’ cancer prevention activity achieved through immunoregulation, the role of immune cells in cancer progression, the role of the mediatory relationship between the gut microbiota and dietary polysaccharides in immunoregulation and cancer prevention are systematically summarized, with the aim of encouraging future research on the use of dietary polysaccharides for cancer prevention.

## 1. Introduction

Polysaccharides, types of natural polymers with ketone groups or aldehyde groups, are formed via the glycosidic bonds of more than 10 monosaccharides. The sources of these polymers are plants, animals and microorganisms. Polysaccharides are types of natural active ingredients widely found in plants with unique biological activity. An increasing number of studies have shown that plant polysaccharides possess antioxidant, anti-tumor, immune stimulation, liver protection, hypoglycemic, gastrointestinal protection and other biological activities. Moreover, due to their advantages of high safety and low side effects, a variety of plant polysaccharides have been widely studied and applied in biochemistry, medicine and the food industry in recent decades. Among them, studies have demonstrated that polysaccharides from *Lycium barbarum* [1], *Ganoderma lucidum* [2] and *Astragalus* [3] possess the ability to enhance immune function, thus facilitating anti-tumor activity, and some of them have been successfully applied in the treatment of cancer.

At present, serious challenges are still met in traditional chemotherapy and radiotherapy for cancer treatment, such as drug resistance and toxic side effects. It is considered that plant polysaccharides have potential value in the development of anticancer agents, since they possess the advantages of multi-pathway, multi-target action; low toxicity; high efficiency; few side effects; and synergistic effects with drugs [4]. Anti-tumor studies of plant polysaccharides mainly focus on liver cancer cells, sarcoma cells, lung cancer cells, human cervical cancer cells, gastric cancer cells, etc. The anti-tumor mechanisms of plant polysaccharides mainly include inhibiting the growth of tumor cells, enhancing immunoregulation and inhibiting the invasion and metastasis of tumor cells. This paper summarizes the sources of dietary plant polysaccharides, the interactions of immune cells and gut microbiota with dietary plant polysaccharides for cancer prevention and the challenges and perspectives for future research of dietary plant polysaccharides. Meanwhile, the structures of plant polysaccharides, including their molecular weight, monosaccharide composition, glycosideric bond type and advanced structure, are closely related to their biological activities, and they are also reviewed in this paper.

## 2. Sources of Dietary Plant Polysaccharides with Immunoregulation

Dietary polysaccharides mainly originate from edible plants, animals, microorganisms and novel food materials [5]. Since the use of polysaccharides from microorganisms for immunoregulation and cancer prevention is systematically reviewed in other articles [6,7,8], in this article, the resources of dietary polysaccharides from edible and medicinal plants with immunoregulatory properties are summarized in Table 1.

### 2.1. Dietary Polysaccharides from Edible Plants

Dietary plant polysaccharides have long been considered the most important sources of polysaccharides. Moreover, these polysaccharides are mainly from cereals, beans, potatoes, fruits, vegetables and algae. Dietary polysaccharides from edible plants account for the most abundant part for polysaccharides, and some of them are active in immunoregulation *in vitro* and *vivo*. Growing research demonstrates that dietary polysaccharides from edible plants, including *Camellia sinensis* (Linn.) O. Kuntze [9], *Undaria pinnatifida* Suringar [10], *Zizania latifolia* (Griseb.) Stapf [11], *Asparagus schoberioides* Kunth [26], *Allium sativum* L. [13], Green algae [27] and Lotus Leaf [28], are effective in immunoregulation and cancer prevention. One of the mechanisms involved in the processes of cancer prevention and immunoregulation is an increase in the secretion of cytokines, which include TNF-α, IFN-γ, IL-1β, IL-2, IL-6, and IL-10 in macrophages. Some of them are also active in regulating the production of NO and ROS and the expressions of iNOS, IL-6 and TNF-α mRNA levels. At the molecular level, the NF-κB and MAPK signaling pathways are regulated in cells by the administration of these polysaccharides.

### 2.2. Dietary Polysaccharides from Edible and Medicinal Plants

In China, many medicinal plants have a long history of use. Among all of them, 110 kinds of food and drug homologous substances have been established in China and 104 kinds from plants to present. Dietary polysaccharides from these edible and herbal plants are regarded as great treasures for human health. Studies have demonstrated that dietary polysaccharides from edible and medicinal plants, such as *Astragalus memeranaceus* [17], *Dendrobium officinale* Kimura et Migo [18], *Panax ginseng* C. A. Meyer [19], *Lycium chinense* Miller [20,21], *Lonicera japonica* Thunb. [22,23], *Atractylodes macrocephala* Koidz [24] and *Rosa laevigata* Michx. [25], are effective in regulating the secretion of cytokines (including TNF-α, IL-1β and IL-6) and NO and in producing ROS in macrophages, which are beneficial for immunoregulation and cancer prevention. Notably, some of them are effective in enhancing the phagocytic function of macrophages and the activity of NK cells to counteract that of cancer cells. Mechanism studies have demonstrated that the activation of the p-JNK MAPK and NF-κB signaling pathways is also involved in the immunoregulatory and cancer prevention processes of these polysaccharides in and ex vitro.

## 3. Effects of Chemical Types of Dietary Plant Polysaccharides on Immunoregulation

### 3.1. Effects of Relative Molecular Mass and Monosaccharide Composition on Immunomodulatory Activity

The immunomodulatory activity of polysaccharides relies on the molecular mass and monosaccharide composition. Reports have demonstrated that polysaccharides with a relatively low molecular mass have difficulty in forming active polymeric structures, and those with too large a molecular mass also have difficulty in crossing the cell membrane into the body to exert their activity [29]. The optimal molecular mass ranges for different polysaccharides exert significant immunomodulatory activities.

The differences in the monosaccharide composition of polysaccharides may result in different immunoregulatory activities; hence, it is of great significance to investigate the relationship between the monosaccharide composition of polysaccharides and their immunoregulatory activities. Chen et al. [30] investigated the *in vitro* immunocompetence of polysaccharides from different developmental stages of shiitake mushroom, and they found that the proportion of galactose and mannose in their monosaccharide composition was the highest at the young mushroom stage, which had the strongest immunocompetence. It was deduced that the proportion of galactose and mannose in their monosaccharides is one of the main factors associated with their higher immunoactivity [30].

Most immunologically active polysaccharides contain galactose, glucose, arabinose, mannose, galacturonic acid, rhamnose, xylose, fucose, fructose, glucuronic acid and so on. For example, the polysaccharides from *Citrus aurantium* L. [31] are composed of galactose, glucose, arabinose and mannose. Results have also demonstrated that immunoregulatory efficacy is correlated with the content of arabinose and glucose in the polysaccharides from this plant. The immune-enhancing activity of polysaccharides from this plant rich in arabinose is significantly stronger than that of other ones. Based on the above-mentioned relationship between the monosaccharide composition of polysaccharides and their immunoregulatory effects, it is speculated that the content of galactose, glucose, arabinose and mannose in polysaccharides directly affects their immunological activity.

### 3.2. Effects of Glycosidic Linkage Forms on the Immunomodulatory Activities of Polysaccharides

Apart from the influence of the monosaccharide composition of polysaccharides on immunological activity, another key factor also affects the activity of polysaccharides. The type of glycosidic bond significantly affects the immunoactivity of polysaccharides. Most polysaccharides with good immunoactivity contain β-(1 → 3) glycosidic bonds. By analyzing the content of β-(1 → 3)-D-glc in polysaccharides from five edible mushrooms, it was found that the immunoregulatory activities of polysaccharides are significantly related to the content of β-(1 → 3)-D-glc [32]. Mannan linked through (1 → 6) glycosidic bonds and galactose linked through (1 → 3) glycosidic bonds for polysaccharides may possess immunological activity, and the polysaccharide drugs currently applied in the clinical treatment of tumors predominately contain β-(1 → 3) glycosidic bonds [33,34].

In another condition, the backbone of polysaccharides is β-(1 → 3) glycosidic bonds, and the side chain is β-(1 → 6) glycosidic bonds, which also shows good immunological activity. Glucans composed of a β-(1 → 3)-D-glucopyranosyl backbone often exert their immunoactivities by recognizing the Dectin-1 receptor. The minimal glucan subunit for Dectin-1 activation is a type of β-(1,3)-D-glucan oligosaccharide with a backbone containing at least seven glucose subunits and a single side chain of β-(1,6)-D-glc at the non-reduction end [33]. Compared with the less branched or linear chains of β-(1 → 3) -d-glucans, the highly branched β-(1 → 3)-d-glucans with β- (1 → 6)-d-glucans as side chains possess higher immunostimulatory activities [34,35], and their activities are gradually enhanced by one side chain branch on every three units of glucose residues along the backbone. This conclusion is also supported by results demonstrating that the immunomodulatory activity on lymphocytes is significantly decreased with the degradation of β-D-1,3-galactosome and d-1,6-galactosyl in polysaccharides from *Astragalus membranaceus* [36]. It has also been indicated that β-D-1,3-galactose and d-1,6-galactosyl share a key role in the immunomodulatory activity of polysaccharides from *Astragalus membranaceus*.

Additionally, the immunomodulatory activity of polysaccharides is correlated with (1 → 2) glycosidic linkages. Results demonstrate that polysaccharides from *Astragalus membranaceus* containing glycosidic bond types of 1 → 2,4-glcp and 1 → 4,6-glcp positively promote the proliferation of B lymphocytes and the secretion of IgG cytokines, which facilitate the enhancement of specific immune function, the phagocytic function of macrophages, the killing activity of NK cells and non-specific immune function [37,38].

In some circumstances, the immunological activity of polysaccharides is related to the structural fragments of specific glycosidic linkages. Results demonstrate that polysaccharides from *Dendrobium officinale* containing fragments of glycosidic bonds →6)-β-D-galp-(1→ and →3,6)-β-D-manp-(1→ are effective in increasing the ratio of CD4^+^ and CD8^+^ T cells, allowing them to exert their cellular immunity and inhibit the proliferation of tumor cells [39,40].

Taken together, it is concluded that the glycosidic linkages between the sugar groups of polysaccharides play a decisive role in the local conformation of polysaccharide molecular chains, which, in turn, affects the spatial morphological structure of the polysaccharides and their immunomodulatory activities. The types and positions of the glycosidic linkages of polysaccharides play an extremely important role in immunoactive effects [41,42].

## 4. The Roles of Immune Cells in Cancer Progression

The progression of cancers is complex and continuously evolving, in which stromal cells, fibroblasts and endothelial cells, as well as innate and adaptive immune cells, are involved and comprise the tumor microenvironment (TME). Hence, it is considered that the regulation of the immune response of TME is beneficial for cancer prevention. The effects of immunoregulation for cancer progression are widely reviewed in other articles; in this review, the roles of the immune cells in TME are briefly summarized in this section. T lymphocytes are particularly focused on for their potent cytotoxic capabilities; hence, in some research, their differentiation status for the “Th1/Th2 paradigm” is considered a model for other cell types [43]. T cells coordinate the pathogen-dependent immune response through the differential production of cytokines: Th1 cells control the pro-inflammatory phenotype, and Th2 cells coordinate the immunosuppressive phenotype.

Dendritic cells (DCs) build the bridge between the adaptive immune system and the innate immune system. DCs initiate the pathogen-specific T-cell response, which is important for enhancing protective immunity. DCs are classified into different subtypes by their functions, including classical DCs (cDCs), plasmacytoid DCs (pDCs) and monocyte-derived inflammatory DCs (moDCs). cDCs are divided into cDC1 and cDC2. The function of cDC1 and cDC2 is under the control of the transcription factors of IRF8, ID2 and BATF3, and of IRF4, ID2, ZEB and Notch2/KLF4, respectively [44]. cDC1s are capable of presenting both endogenous (the antigens of tumors) and exogenous antigens, whereas cDC2s only present exogenous antigens to T cells.

Neutrophils, the first line of defense against pathogens, account for 70% of circulating leukocytes [45]. Usually when tissue is damaged or infected, epithelial cells secrete neutrophil-homing chemokines to force neutrophils to seep out of the circulation and enter the damaged tissue, where neutrophils secrete inflammatory cytokines and release neutrophil extracellular traps (NETs) for the phagocytosis of invasive microorganisms [46]. Tumor-associated neutrophils (TANs) also follow the Th1/Th2 paradigm under the conditions of cancer, showing N1 (tumor suppression) or N2 (tumor promotion) phenotypes. The phenotype of the neutrophils in TME is often determined by the type of tumor and the stage of disease progression. Neutrophils exert inflammatory effects in the early stage of tumors; however, they transform into the immunosuppressive phenotype as the tumor progresses [47]. In this stage, neutrophil elastase (NE) and matrix metalloproteinase (MMP8/9) are secreted for the reconfiguration of the extracellular matrix in TME, which is beneficial for promoting tumor angiogenesis (Oncostatin-M), tumor progression (PGE2) and invasion (through the release of ROS/RNS, NE and MMP-9) [48,49].

Natural killer cells (NKs) are a category of circulating congenital lymphocytes, which are well known for their cytotoxic effector functions. The NK subgroups are defined by the expressions of CD16 and CD56 levels, that is, CD56^hi^ CD16^+/−^ and CD56^lo^ CD16^hi^ [50]. The biological divisions of the two subsets are significantly different: CD56^hi^ CD16^+/−^ NKs secrete inflammatory cytokines, whereas CD56^lo^ CD16^hi^ NKs specialize in cytotoxic functions and cell-mediated killing. In the background of cancer, these cells are very effective in eliminating malignant cells and limiting tumor metastasis [51]. NKs often induce death-receptor-mediated apoptosis and perforin/granzyme-mediated cytotoxicity to target tumor cells and limit the growth of primary tumors [52]. Although NKs are characterized by destroying circulating tumor cells, their cell killing efficiency in TME is much lower.

Another important component of the TME is the innate lymphoid cells (ILCs), which share the same characteristics as NK cells. There are three known phenotypes of ILCs (ILC1, ILC2 and ILC3), which are classified by their production of Th1-, Th2- and Th17-based cytokines and different transcription factors [53]. ILC1s contribute their anti-tumor effects by producing cytokines (mainly IFN-γ). ILC3s are typical tumor-promoting cells, but ILC2s either promote or antagonize tumor growth, which is determined by tumor type [54]. A representative scheme of immune cells for cancer prevention is presented in Figure 1.

## 5. Mechanisms of Dietary Plant Polysaccharides for Cancer Prevention and Immunoregulation

The modulatory effects of dietary plant polysaccharides on the immune system are mainly mediated by non-specific or specific immune enhancement and stimulation. The main receptors for dietary plant polysaccharides reported in the current literature are TLRs, Dectin-1 receptors, complement receptors, scavenger receptors, etc. [55,56]. Upon binding to the receptors, the intracellular pathways of immune cells are activated by polysaccharides, triggering the release of cytokines to exert their immunoactivities. The binding of polysaccharides with different cell receptors triggers different intracellular signal transduction pathways; for example, the serine-activated immune receptor recognition region of the type II transmembrane receptor in the cytoplasm is activated by polysaccharides binding to Dectin-1 receptors, which further triggers the production of nitric oxide and the secretion of cytokines, affecting the activity of intracellular enzymes and reducing the secretion of the migration factors of macrophages [57,58]. In other conditions, when polysaccharides bind to Toll-like receptors, cellular immunity is enhanced via the activation of the intracellular TLR4 signaling pathways of macrophages [59]. With the help of Th cells and the secretion of IL-10 cytokines, phagocytes can effectively present the antigens to the B lymphocytes and stimulate the secretion of IgG and IgM antibodies to recognize and bind to the exogenous antigens on tumors. Meanwhile, Th cells can activate Tc cells mediated by IL-12 cytokines, which facilitate the release of INF cytokines to exert their immunoregulatory effects and prevent tumor growth. In some circumstances, polysaccharides exert their immunomodulatory effects by directly inhibiting the activities of chemokine CCL5 and pro-inflammatory factor IL-6, or by enhancing immunoactivities by promoting the secretion of colony-stimulating factors for the proliferation of cells [60]. Research reports that polysaccharides from *Saliva chinensis* Benth. exert their immunoregulatory effects by activating the PTEN pathway to increase the ratio of CD4^+^ and CD3^+^ T cells in peripheral blood, promote the protein expression of PTEN and inhibit the expressions of PI3K and p-Akt [61]. Polysaccharides from Coix seeds are effective in activating the AK3/STAT5 signaling pathway mediated by IL-2, which promotes the proliferation of T lymphocytes and allows them to exert their immunological activities [62]. Due to the intrinsic difference in the distribution of receptors on the surface of immune cell membranes and the decisive role of polysaccharide binding with receptors for cellular transduction, the mechanisms of dietary plant polysaccharides in cancer prevention mediated via interactions with immune cells are detailed in the following section.

### 5.1. Regulation of the Immunoactivities of Monocytes and Macrophages

#### 5.1.1. Induction of M1 Phenotype Polarization of Macrophages

There are two different functional phenotypes of macrophages, and they are M1 and M2 phenotype macrophages. In particular, M1 macrophages possess anti-pathogen and anti-tumor activities. Research has demonstrated that the activation of Notch signaling pathways is correlated with the upregulated gene expression of M1 phenotype polarization. Meanwhile, the activation of the Notch signaling pathways in macrophages also promotes the oxidative phosphorylation of glucoses and the production of reactive oxygen species (ROS) to further stimulate the expressions of M1-type polarized genes [63]. In research on the interactions between polysaccharides from *Astragalus membranaceus* and macrophages, the results demonstrate that polysaccharides from *Astragalus membranaceus* are effective in upregulating the expression levels of M1 gene markers, and mechanism studies show that Notch signaling pathways are involved in the induction of macrophage polarization to the M1 phenotype for polysaccharides from *Astragalus membranaceus* [64]. Research also demonstrates that some dietary polysaccharides are effective in inhibiting or enhancing the phagocytic activities of macrophages and neutrophils by regulating the release of acid phosphatase and NO [65]. Some dietary plant polysaccharides are effective in decreasing the production of ROS in human neutrophils by stimulating the inducible nitric oxide synthase protein expression of macrophages [66].

#### 5.1.2. Induction of Dendritic Cell (DC) Maturation and Activation

DCs, the main antigen-presenting cells, play an important role in the immune response by activating helper T cells (HTCs) and cytotoxic T cells. Polysaccharides derived from plants are also effective in stimulating the maturation of DCs, which possess the ability to present the internalized tumorigenic antigens to naive T cells and subsequently prime T cells for tumor elimination. Research demonstrates that polysaccharides from *Portulaca oleracea* L. are effective in the prevention of tumor growth by inducing the maturation of DCs and increasing the protein expression levels of CD80 in tumor cells [67]. Other research results also demonstrate that polysaccharides from *Lycium barbarum* possess the ability to stimulate the maturation of DCs, which facilitates the enhancement of the immune response and anti-tumor activity [21,68].

### 5.2. Regulation of the Immunoactivities of Lymphocyte Subsets

Among all the lymphocyte subsets, natural killer (NK) cells are highly capable of clearing viruses and tumors. Additionally, the differentiation, activation and recruitment of other types of immune cells are also stimulated by the secretion of cytokines, such as IFN-γ released by NK cells, which are beneficial for both innate and adaptive immunity [69,70]. A research study on the polysaccharide fraction (GS-P) from the leaves of *Panax ginseng* demonstrated that the secretion of TNF-α and interleukin IL-12 was enhanced in the peritoneal exudate macrophages (PEMs) of GS-P-treated mice, and it also found that the cytotoxicity of PEMs and NK cells isolated from GS-P-treated mice against colon 26-M3.1 cells and lymphoma cells (YAC-1) was significantly enhanced. Moreover, the inhibitory effects of GS-P on lung metastasis were also validated in their experiments, demonstrating that GS-P exhibits antimetastatic activity by promoting the activation of macrophages and NK cells [71].

Plant polysaccharides can also improve the body’s immunity by promoting the proliferation of lymphocytes. For example, research has demonstrated that polysaccharides from *Dioscorea polystachya* Turczaninow are effective in promoting the proliferation of T lymphocytes and enhancing NK cell activity, thus exerting their regulatory effects on the body’s immunity [72,73]. It has also been reported that polysaccharides from *Asparagus schoberioides* Kunth have novel immunomodulatory effects when stimulating the proliferation of spleen lymphocytes in mice [74]. In counteraction with tumors, T helper cells (Ths) also play a crucial role in this process by secreting a variety of cytokines. Among them, Th1 cells mediate cellular immunity, and Th2 cells mediate humoral immunity; the ratio of Th1/Th2 always maintains a dynamic balance in the body under normal conditions. However, this dynamic balance will be broken once the body is invaded by exogenous or endogenous pathogens, for example, viruses or parasitic infections, tumors and so on, which induce the immune drift of the body. Plant polysaccharides possess the ability to regulate this dynamic balance in the body, thus exerting their cancer prevention effects. Polysaccharides from radix glycyrrhizae have been demonstrated to be effective in suppressing the growth of tumor cells through the downregulation of Treg cells and the upregulation of the Th1/Th2 cytokine ratio in plasma in H22-hepatocarcinoma-bearing mice [75]. Additionally, the expressions of Th1/Th2-related genes are also regulated by polysaccharides from *Angelica sinensis,* which is also effective in enhancing body immunity and for cancer prevention [76]. Ginseng polysaccharide injections, used as adjuvants for the clinical treatment of non-small-cell lung cancer (NSCLC), are effective in improving the therapeutic effects of chemical drugs and the quality of patients’ lives [77]. A mechanism study demonstrated the successful restoration of the immune drift induced by the imbalance of Th1/Th2 in patients [78]. A recent study found that the natural source of glycans from plants, Man9GlcNAc2Asn, exhibits high-affinity interactions with C-type lectin receptors (CLRs) and might be involved in a cytotoxic T-lymphocyte immune response mediated by CD8+ T cells [79], which showed the great significance of this kind of plant polysaccharide for the development of new targeted therapies.

## 6. Relationship between Gut Microbiota and Dietary Plant Polysaccharides and Immunoregulation in *In Vivo* Cancer Prevention

### 6.1. Interactions between Gut Microbiota and Dietary Plant Polysaccharides

Gut microbiota play a critical role in immunoregulation and in maintaining intestinal health. In recent years, more and more have studies focused on the regulatory role of dietary plant polysaccharides in gut microbiota. Gut microbiota, a highly complex microbial community in the gastrointestinal tract, are often involved in the material metabolism, absorption and synthesis of nutrients, promoting growth and development, as well as maintaining the normal physiological activities of the human body [80,81]. Increasing evidence demonstrates that the imbalance of gut microbiota is closely related to the development of enteritis, intestinal cancer and other diseases [81]. Plant polysaccharides alleviate intestinal diseases mainly by regulating the structure and metabolism of gut microbiota, which also facilitates an improvement in intestinal immunity by maintaining the integrity of the intestinal barrier [82]. Plant polysaccharides, as biological macromolecules, are not easily digested or absorbed by the body. However, they can be used as the carbon source of gut microbiota, with fermentation in the large intestine occurring to exert the effects of anti-inflammation, protection of the intestinal mucosa and intestinal immunity regulation by affecting the structure of the intestinal flora and its metabolites [83,84]. Plant polysaccharides are often metabolized by intestinal microorganisms and produce a series of products, such as short-chain fat acids (SCFAs), indole derivatives and polyamines, which directly regulate intestinal homeostasis and disease progression [85,86]. Among all the metabolites, SCFAs are widely considered as being closely related to intestinal energy supply, the maintenance of the homeostasis of the intestinal mucosal barrier, the regulation of intestinal motility, and intestinal immunity and anti-tumor activity [87,88,89]. Liu et al. found that *Astragalus* polysaccharide treatment is beneficial for relieving the constipation symptoms of rats by reducing the production of acetate, butyrate and propionate in the rats’ feces and by regulating glycolysis/gluconeogenesis and pyruvate metabolism [90]. Research results from Yang et al. demonstrate that the administration of turmeric polysaccharides significantly improves the pathological phenotype and reduces the damage to the intestinal barrier in mice with colitis with an increased level of indole-3-acetic acid and its ligand aromatic receptor, as well as the metabolites of tryptophan in the caecum [91]. A study conducted by Fu et al. demonstrates that purslane polysaccharides are effective in regulating the abnormalities of blood lipids in aging rats by reducing the ratio of thick-walled bacteria/bacteroides and the relative abundance of Fusobacterium in the intestine of the rats [92]. Polysaccharides from *Lycium barbarum* are proved to be effective in regulating the composition of intestinal flora and the metabolism of SCFAs in rats with obesity [93]. Results from the simulated gastrointestinal digestion and fecal fermentation of polysaccharides from Rhizoma Dioscoreae demonstrate that the free glucose and mannose digested from polysaccharides and utilized by intestinal flora are increased, and the production of SCFAs is upregulated, which is deemed to correlate with the novel *in vivo* anti-inflammatory effect [94].

### 6.2. Interactions between Gut Microbiota and Immune System for Cancer Prevention

A huge number of microbial communities colonize the human gut. Due to the existence of intestinal secretions and mucosal tissue, the intestinal flora and human body form a relatively isolated state, called the epithelial barrier, and, hence, the interactions between the gut microbiota and the human body are often ignored. However, the intestinal microflora sometimes translocates to adjacent organs of the intestine or participates in the progression and development of tumors by regulating the metabolism of the human body [94,95,96]. Statistically, about 20% of malignant tumors are related to microorganisms (mainly related to bacteria), and 99% of them colonize the human gut. The role and mechanisms of gut microbiota in immunoregulation and cancer prevention are presented in the following section.

#### 6.2.1. Gut Microbiota Affects Tumor Progression by Regulating Body Immunity

The intestinal flora is an effective stimulator of the intestinal immune response, and it is related to the dynamics of human immune cell reconstruction. It has been observed that the recovery rate of neutrophils is upregulated with an increase in and high abundance of Fecal bacilli, Ruminococcus and AKK bacteria. Meanwhile, the recovery of neutrophils is decreased by an increased abundance of Rochella and Clostridium in the gut of patients receiving hematopoietic stem cell transplantation. The intestinal flora also significantly affects the reconstruction of lymphocytes and monocytes [97]. Smruti et al. found that abnormal flora can induce innate and specific immune tolerance through the effect of Toll-like receptors on the surface of monocytes and provide a beneficial immune microenvironment, including a reduction in inhibitory cells from bone marrow, the inhibition of M1 macrophages and killer T-cell differentiation, which is conducive to pancreatic cancer cell growth [98]. In an experiment, Bifidobacterium pseudolongum was found to promote the development of pancreatic cancer. Similarly, Berk et al. found that the intestinal flora of mice with pancreatic cancer is dysfunctional, and a portion of the flora translocates to the diseased pancreas [95]. This dysfunctional flora (especially for the significantly increased Malassezia) activates the complement cascade reaction through the MBL pathway, and the cascade products, C3a, bind to its receptors to promote tumor development [95]. It has been demonstrated that the anti-tumor effects of the human body cannot be separated from the participation of intestinal bacteria. Research results show that Akkermansia muciniphila enhances the anti-tumor effect of immune checkpoint inhibitors designed for PD-1 and CTLA-4 [99]. Takeshi et al. also found that the implantation of the combination of eleven healthy bacterial strains in human intestines induces the differentiation of CD8 T cells (mainly the secretion of IFN-γ) and enhances the inhibition effects of immune checkpoint inhibitors for tumor growth [100]. Upon the occurrence of colitis, the intestinal microflora is significantly changed, which is accompanied by the activation of TLR4 of epithelial cells and upregulation of the expression of CCL2 in epithelial cells caused by lipopolysaccharides from abnormal microflora. The high expression of CCL2 induces the accumulation of mononuclear macrophages in the colon and promotes the development of colorectal cancer. At the same time, lipopolysaccharides induce an increased expression of IL-1β in monocyte-like macrophages, which promotes the expansion of Th17 cells and forms a vicious circle between intestinal microbial imbalance and inflammation [101].

#### 6.2.2. Gut Microbiota Affects Tumor Progression by Regulating Body Metabolism

The intestinal flora, colonizing the gut, affect the circadian rhythm and metabolism of the host by producing metabolites such as short-chain fatty acids. An abnormal intestinal flora may lead to the risk of metabolic syndrome and increase the possibility of cancer in humans [80]. The liver is one of the organs mostly affected by intestinal microflora metabolites since about 70% of the blood in the liver originates from enterohepatic circulation. The accumulation of CXCR6^+^ NKT cells in the liver (positively correlated with the expression of CXCL16 on the hepatic sinus endothelial cells) is effective in inhibiting hepatocarcinogenesis. Primary bile acids promote the expression of CXCL16, while secondary bile acids have the opposite effects. Research shows that Clostridium can promote tumor development by promoting the conversion of primary bile acids into secondary bile acids [96]. CXCL16 is also found in the exosomes of colorectal cancer cells infected with Fusobacterium nucleatum, and research demonstrates that this kind of exosome can be transferred from tumor cells into normal cells to promote the metastasis of colorectal cancers [102]. In the case of intestinal inflammation, intestinal microorganisms interact with the intestinal epithelial cells on the damaged intestinal barrier, which leads to a decreased content of all trans retinoic acids and the secretion of CD8^+^ T cells in the tumor microenvironment via the regulation of the activities of all-trans retinoic acid synthase and catabolic enzymes, and this also accelerates the growth of colorectal cancer cells [103].

The intestinal flora often metabolizes tryptophan in foods into a variety of indole derivatives, D-tryptophan and other products. These metabolites display the activities of intestinal inflammation regulation, mucosal immunity and intestinal barrier function. Through the regulation of microbial metabolism, intestinal inflammation and related cancer can be prevented. Indole propionic acid is one of the metabolites of tryptophan, which inhibits the proliferation and metastasis of mouse breast cancer cells. In the early stage of breast cancer, intestinal microbial indole synthesis is often inhibited, which indicates that the intestinal flora may become a new target for the prevention and therapy of breast cancer. Mutation of the P53 gene is recognized as a common cause of cancer. Eliran et al. found that the mutant P53 gene exhibits a tumorigenic effect in the distal intestine, while it exerts a tumor inhibition effect in the proximal intestine [104]. A study demonstrated that one of the intestinal flora metabolites, gallic acid, plays a key role in the promotion of tumor development by regulating the expression of the mutant P53 gene in the distal intestine [104].

## 7. Future Research Challenges and Perspectives

Taken together, an abundancy of research suggests that dietary plant polysaccharides have obvious activities in cancer prevention; however, there remain some challenges and difficulties in determining the underlying mechanisms of polysaccharides for cancer prevention. First, polysaccharides, a type of compound with a complex composition, a diversity of chemical structures and differences in physical and chemical properties, display many disadvantageous factors in in-depth studies on them. The complexity of polysaccharides also results in considerable difficulties in studying their effects on immunoregulation and cancer cell inhibition. Taking this issue into consideration, in-depth research work might focus on the correlation between the chemical structure of active polysaccharides and their immunoactivity, cancer prevention in the future and the establishment of the dynamic relationship between the chemical profile of polysaccharides and the corresponding effects. Meanwhile, with the help of advanced analysis and separation technologies in the analysis of polysaccharides, future work might pay more attention to the relationship between the composition information of polysaccharides and their biological activities. Second, due to the special properties of polysaccharides, it is too difficult for them to cross through biological barriers to play a direct regulatory role after oral administration. Therefore, it is often believed that polysaccharides play an indirect role in immunoregulation and cancer prevention *in vivo*, which is greatly dependent on the mediators, including microorganisms and their metabolites. As one of the hosts of the human body, the composition of intestinal microorganisms is also very complex. Research on the regulatory effect of polysaccharides on microbiota is gradually increasing, but the potential mechanisms of polysaccharides in the regulation of microbiota still need to be determined. For example, the correlations between regulation activities and different types, structures and compositions of polysaccharides and classes of microbiota need to be identified. Moreover, determining whether the regulatory effect of polysaccharides on microbiota is a direct action effecting the growth of them, or whether it works by regulating the metabolites of specific microorganisms, is also important in better understanding this important biological process. The changes in metabolites caused by disturbance to the microbial composition of the homeostasis of human immune systems, as well as the response of the human body to these changes and so on, are deserving of further consideration. In addition, most of the current research on the effects of polysaccharides on immunoregulation is performed on animals under pathological conditions. However, researchers should also pay attention to the impact of bioactive dietary polysaccharides on immune systems and intestinal microecosystems under healthy conditions, with the aim of providing sufficient evidence for the safety of polysaccharides. Moreover, such research might be helpful for clarifying the potential function of the healthcare or therapeutic effects of these active polysaccharides.

In short, the study of the use of dietary polysaccharides for the regulation of immune systems and cancer prevention will be a long and arduous task. In this research process, the human body can be regarded as a complex biological system, and the effect on the immune systems is the key point of the research, which includes the differentiation, activation, maturation and transformation of immune cells and so on. The intestinal microflora that directly interacts with polysaccharides is regarded as another complex biological system, and research on this mainly focuses on the impacts of polysaccharides on the composition and dynamic changes in microorganisms and the transmission of message materials during the process of intestinal microflora interacting with polysaccharides. Regarding dietary polysaccharides, their composition is complex, and their structural diversity can be regarded as another complex system. We should focus on the correlations between the chemical structure and biological activity, especially for the relationship between the chemical structure of polysaccharides and the effects of regulation on micro-ecology. Only by clarifying these complex issues can we better understand the potential mechanisms behind polysaccharides exerting their immunoregulatory and cancer prevention effects, which will also allow us to build a more solid foundation for the in-depth development of polysaccharides with better activities for healthcare or therapeutic services. We have reason to believe that, with interdisciplinary integration in the future, such as the application of the emerging technologies in the field of research on life sciences, including the combination of multiomics technologies, artificial intelligence and big data analysis, we will provide better technical support and the stronger development necessary for revealing these complex issues.

## Figures and Tables

**Figure 1 nutrients-15-03019-f001:**
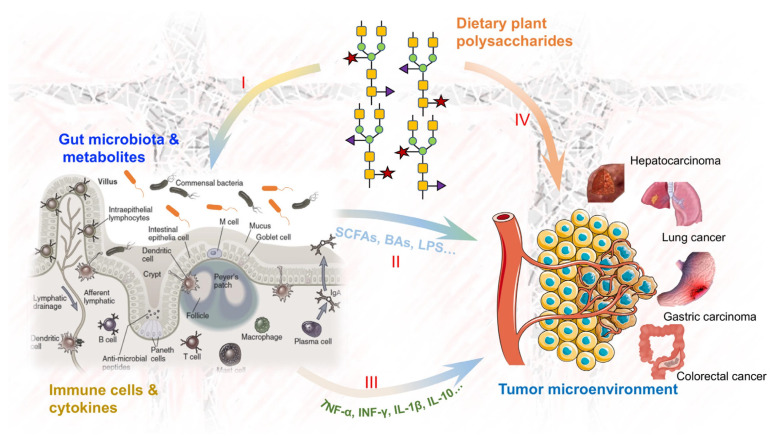
Scheme of dietary plant polysaccharide interaction with gut microbiota and immune cells for cancer prevention (I, dietary plant polysaccharides with mass molecules are often not easily absorbed by the human body, and they are often used as important materials for gut microbes and are widely metabolized into small molecules; II, some metabolites (such as SCFAs, BAs, LPS and indoles) produced by gut microbes directly interact with the immune cells in the gut or are released into the blood to indirectly exert their immunoregulatory activity; III, some cytokines (such as TNF-α, INF-γ, IL-1β and IL-10) secreted by the gut immune cells transfer to the blood to exert their immunoregulatory and cancer prevention activities; IV, evidence also supports the notion that some bioactive dietary plant polysaccharides with small molecules are effective in immunoregulation and cancer prevention, and this is also considered another one of their actions).

**Table 1 nutrients-15-03019-t001:** Summary of the main sources of dietary polysaccharides from plants with immunoregulatory and cancer prevention properties.

Category	Polysaccharides from Plants	Mechanisms for Immunoregulation	Reference
Edible plants	*Camellia sinensis* (Linn.) O. Kuntze	Increase in TNF-α, IFN-γ, IL-1β, IL-2 and IL-6 levels in serum	[9]
	*Undaria pinnatifida* Suringar	Activation of Toll-like receptors (TLRs), NF-κB and other immune-related signaling pathways	[10]
	*Zizania latifolia* (Griseb.) Stapf	Enhancement of the ability of phagocytosis and NO production of macrophages	[11]
	*Gracilaria lemaneiformis*	Activation of the expressions of iNOS, IL-6 and TNF-α mRNA; enhancement of the proliferation and pinocytosis of macrophages; promotion of the production of ROS, NO, IL-6 and TNF-α	[12]
	*Allium sativum* L.	Stimulation of NO release by macrophages	[13]
	*Chlorella ellipsoidea*	Stimulation of mouse macrophages to produce NO and various cytokines (IL-1, IL-6, IL-10 and IL-12); activation of the NF-κB and MAPK pathways	[14]
	*Nelumbo nucifera* Leaves	Activation of mRNA expression of cell proliferation and cytokine secretion; activation of MAPK and NF-κB pathways	[15]
	*Lactuca sativa* Linn.	Promotion of macrophage proliferation, phagocytosis and NO production	[16]
Edible and medicinal plants	*Astragalus memeranaceus*	Enhancement of phagocytic function of macrophages and activity of NK cells; promotion of the function of non-specific and specific immune responses	[17]
	*Dendrobium officinale* Kimura et Migo	Enhancement of proliferative activity, swallowing activity, NO release and ROS production of macrophages	[18]
	*Panax ginseng* C. A. Meyer	Promotion of the secretion of cytokines in macrophages; activation of the corresponding signaling pathways	[18,19]
	*Lycium barbarum* L.	Regulation of the production of NO, ROS, TNF-α, IL-6 and IL-1β in macrophages; promotion of the proliferation and phagocytic capacity of macrophages; elevation of the mRNA expression of iNOS; and activation of p-JNK MAPK signaling pathway	[20,21]
	*Lonicera japonica* Thunb.	Improvement of the immunosuppression index of immune organs and promotion of the proliferation and cytokine release of immune cells	[22,23]
	*Atractylodes macrocephala* Koidz.	Promotion of the proliferation and NO release in macrophages	[24]
	*Rosa laevigata* Michx.	Enhancement of phagocytosis, secretion of cytokines (TNF-α, IL-6 and NO) and mRNA expression; activation of MAPKs and NF-κB signaling pathways	[25]

## Data Availability

Not applicable.

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
