# Peer review of "Dietary Plant Polysaccharides for Cancer Prevention: Role of Immune Cells and Gut Microbiota, Challenges and Perspectives"

_nutrients, 2023, doi:10.3390/nu15133019_

Round 1
Reviewer 1 Report
Very interesting and well organized. While presenting less known findings associated with plant polysaccharides, the article does not present much depth with how the plant compounds interact with host immune cells.
The large section 4 summarizes immune cell function. This content has been reviewed in hundreds of other reviews and does not provide additional context for the understanding of plant polysaccharide function. This is especially true since there are brief but sufficient summaries of immune cell functions in section 5.1
Similarly, there is a huge discussion of intestinal flora that is extraneous to the emphasis of the role of plant polysaccharides (section 6.2.1).
Overall, the review lacks depth in its description of plant polysaccharides as modulators of immune cell function. It very loosely presents a basis for a connection among plant polysaccharides, immune cell function, gut microbiota and cancer.
The article is readable but needs moderate grammatical review.
Example: Lines 135-7.
Authors use the "researches" in the plural to describe the collection of research studies. It is acceptable that "research" can be used in the plural.
Example "Much research shows...." rather than "Much researches show".
Author Response
Reviewer 1#:
Comments and Suggestions for Authors
Q1: Very interesting and well organized. While presenting less known findings associated with plant polysaccharides, the article does not present much depth with how the plant compounds interact with host immune cells.
A1: Many thanks for your comments. It is true that there are less depth findings for the plant polysaccharides interact with host immune cells in our previous article. For one thing, it is a big challenge for researching on plant polysaccharides interact with host immune cells since the plant polysaccharides are often widely metabolized in the gastrointestinal tracts, and merely direct interact with host immune cells, for another thing, it is found most researches mainly focus on the effects of plant polysaccharides for regulation of the release of cytokines by immune cells but not report the direct interactions between plant polysaccharides and host immune cells. In the future research work, more evidence on this topic should be specifically elaborated and validated. We also add some additional research results in our revised manuscript according to your comments.
Q2: The large section 4 summarizes immune cell function. This content has been reviewed in hundreds of other reviews and does not provide additional context for the understanding of plant polysaccharide function. This is especially true since there are brief but sufficient summaries of immune cell functions in section 5.1
A2: Thanks for your constructive comments. It is true that function of the immune cells for cancer progression are summarized in section 4. We attempt to emphasize the significance of immune systems for the cancer prevention and lay a foundation for reviewing on the role of plant polysaccharides for immune regulation in the following section. It is thought the direct anticancer activity of plant polysaccharides is less but the possibility of regulation of immune system by plant polysaccharides for cancer prevention might be great. Hence, in this article the role of the immune cells for cancer progression are specifically reviewed in this section, and the role of plant polysaccharides for regulation of different kinds of immune cells are separately reviewed in the following section.
Q3: Similarly, there is a huge discussion of intestinal flora that is extraneous to the emphasis of the role of plant polysaccharides (section 6.2.1).
A3: Thanks for your comments. It is believed the intestinal flora are widely distribute in human digestive organs, and they can be considered as another important human organ due to their huge amount and important role for human life and health. In this article, the role of intestinal flora is discussed mainly regarding on the following considerations: I, the intestinal flora directly participates in the digestion of plant polysaccharides, and there exists a direct interaction between composition of intestinal flora and plant polysaccharides; II, different composition of intestinal flora may lead to the production of significant different small molecular metabolites and induce the significance different immune response in human. Hence, the intestinal flora is regarded as an important bridge building for plant polysaccharides and human immune systems, and the role of intestinal flora are summarized in this section.
Q4: Overall, the review lacks depth in its description of plant polysaccharides as modulators of immune cell function. It very loosely presents a basis for a connection among plant polysaccharides, immune cell function, gut microbiota and cancer.
A4: Thanks for your comments. It is true to clarify the exact connection among plant polysaccharides, immune cell function, gut microbiota and cancer, since all of them can be considered as complex systems. As discussed in the article, polysaccharides, a kind of compounds in complex composition, diversity of chemical structures, differences of physical and chemical properties, bring many disadvantageous factors for the in-depth study on them. As one of the hosts of human body, the composition of intestinal microorganisms is also very complex. Researches on the regulation of polysaccharides on microbiota are gradually increasing, but the potential mechanisms of polysaccharides for regulation of microbiota are still need to be disclosed. The human immune system is also very complex and accurately regulated by many factors and cytokines. Hence, it is still a big challenge for investigators to draw a clear clue for the connection between plant polysaccharides, immune cell function and gut microbiota for cancer prevention. In future, it is believed big progress might be made for providing evidences for the connection between plant polysaccharides, immune cell function and gut microbiota for cancer prevention with the help of modern advanced technologies for research of life sciences.
Q5: Comments on the Quality of English Language: the article is readable but needs moderate grammatical review. Example: Lines 135-7. Authors use the "researches" in the plural to describe the collection of research studies. It is acceptable that "research" can be used in the plural. Example "Much research shows...." rather than "Much researches show".
A5: Thanks for your kind remind and comments. The language grammar of the sentences is all reviewed and revised.
Reviewer 2 Report

Moderate editing of English language is required.
Author Response
Reviewer 2#:
This manuscript described regulation of immune cells and gut microbiota by dietary plant polysaccharides for cancer prevention. The topic is interested. However, there are some issues to consider.
Table1
Q1: Increase the TNF-α, IFN-γ, IL-1β, IL-2 and IL-6 levels in serum
I suggest to revise as follows: Increment of TNF-α, IFN-γ, IL-1β, IL-2 and IL-6 levels in serum.
A1: Thanks. Accepted and revised.
Q2: Activate toll-like receptors, NF-κB and other immune-related signal pathwayS
I suggest to revise as follows: Activation of toll-like receptors (TLRs), NF-κB and other immune-related signal pathways
A2: Thanks. Accepted and revised.
Q3: Enhance the ability of phagocytosis and NO production of macrophages I suggest to revise as follows: Enhancement of the ability of phagocytosis and NO production of macrophages
A3: Thanks. Accepted and revised.
Q4: Activation of the expression of iNOS, IL-6 and TNF-α mRNA; enhance the proliferation and pinocytosis of macrophages, prompting the production of ROS, NO, IL-6 and TNF-α
I suggest to revise as follows: Activation of the expression of iNOS, IL-6 and TNF-α mRNA; enhancement of the proliferation and pinocytosis of macrophages, promotion of ROS, NO, IL-6 and TNF-α. production.
A4: Thanks. Accepted and revised.
Q5: Stimulating mouse macrophages to produce NO and various cytokines (IL-1, IL-6, IL-10, and IL-12)
I suggest to revise as follows: Stimulation of mouse macrophages to produce NO and various cytokines (IL-1, IL-6, IL- 10, and IL-12)
A5: Thanks. Accepted and revised.
Q6: Activate mRNA expression of cell proliferation and cytokine secretion; activate MAPK and NF-κB pathways
I suggest to revise as follows: Activation of mRNA expression of cell proliferation; cytokine secretion; MAPK and NF-κB pathways activation.
A6: Thanks. Accepted and revised.
Q7: Promote macrophage proliferation, phagocytosis and NO production I suggest to revise as follows: Promotion of macrophage proliferation, phagocytosis and NO production
A7: Thanks. Accepted and revised.
Q8: Enhance phagocytic function of macrophages and activity of NK cells; promote the function of non-specific and specific immune
I suggest to revise as follows: Enhancement of phagocytic function of macrophages and activity of NK cells; promotion of the function of non-specific and specific immune
A8: Thanks. Accepted and revised.
Q9: Enhance proliferative activity, swallowing activity, NO release, and ROS production of macrophages
I suggest to revise as follows: Enhancement of proliferative activity, swallowing activity, NO release, and ROS production of macrophages
A9: Thanks. Accepted and revised.
Q10: Promotes the secretion of cytokines in macrophages; activate the corresponding signaling pathways
I suggest to revise as follows: Promotion of cytokines secretion in macrophages; activation of the corresponding signaling pathways
A10: Thanks. Accepted and revised.
Q11: promote proliferation, phagocytic capacity of macrophage; elevate the mRNA expression of iNOS and activate p-JNK MAPKs signaling pathway
I suggest to revise as follows: Promotion of proliferation, phagocytic capacity of macrophages; elevation of the mRNA expression of iNOS and activation JNK MAPKs signaling pathway
A11: Thanks. Accepted and revised.
Q12: Improve the immunosuppression index of immune organ and promote the proliferation and cytokine release of immune cells
I suggest to revise as follows: Improvement of the immunosuppression index of immune organ and promotion of the proliferation and cytokine release of immune cell
A12: Thanks. Accepted and revised.
Q13: Promote the proliferation and NO release in macrophage I suggest to revise as follows: Promotion of the proliferation and NO release in macrophages
Enhance phagocytosis, secretion of cytokines (TNF-α、IL-6 and NO) and mRNA expression, activate MAPKs and NF-κB signaling pathways I suggest to revise as follows: Enhancement of phagocytosis, cytokines secretion (TNF- α、IL-6 and NO), mRNA expression, MAPKs and NF-κB signaling pathways activation
A13: Thanks. Accepted and revised.
Q14: No references were cited in section 2.1 and 2.2. Line 102
Mechanism “studies” demonstrate.....
A14: Thanks. Accepted and revised.
Q15: Line 213, 214, 215 CD56hi CD16 +/- and CD56lo CD16hi “hi”, “+/1” and “lo” should be superscripted.
A15: Thanks. Accepted and revised.
Q16: Line 307 ...of tumor necrosis factor (TNF) -α and interleukin (IL)-12... Should revise as follow: ...of TNF-α and IL-12.....
A16: Thanks. Accepted and revised.
Q17: Line 413 Toll-like receptor 4 of.... Should revise as follow: TLR4 of....
A17: Thanks. Accepted and revised.
Q18: Line 416 TH17 should revise as “Th17”.
A18: Thanks. Accepted and revised.
Q19: The role of tumor-associated neutrophils (TAN) in cancer progression were mentioned in paragraph 4 “the role of immune cells for cancer progression”. However, few discussions of dietary plant polysaccharides for TAN regulation in later paragraph of the manuscript.
A19: Thanks for your kind remind and comments. The discussions of dietary plant polysaccharides for TAN regulation are added in paragraph 5.
Round 2
Reviewer 2 Report
n I did not find any changes in section 2.1 and 2.2. No references were cited in section 2.1 and 2.2.
n Table 1
…. Activation of toll-like receptors, NF-κB and other immune-related…
Should revise as follow:
….. Activation of toll-like receptors (TLRs), NF-κB and other immune-related…
n Line 226
..are toll-like receptors, Dectin-1 receptors….
Should revise as follow:
…are TLRs, Dectin-1….
n Line 404
TH17 should revise as “Th17”.
Author Response
Q1: I did not find any changes in section 2.1 and 2.2. No references were cited in section 2.1 and 2.2.
A1: Thanks for your kind remind. The references were correctly cited in section 2.1 and 2.2.
Q2: Table 1 .... Activation of toll-like receptors, NF-κB and other immune-related... Should revise as follow: ..... Activation of toll-like receptors (TLRs), NF-κB and other immune-related...
A2: Accepted with many thanks.
Q3: Line 226 ..are toll-like receptors, Dectin-1 receptors.... Should revise as follow: ...are TLRs, Dectin-1....
A3: Accepted with many thanks.
Q4: Line 404 TH17 should revise as “Th17”.
A4: Accepted with many thanks.